# Next of kin participation in the care of older persons in nursing homes: A pre–post non-randomised educational evaluation, using within-group and individual person-level comparisons

**Albert Westergren**[1,2]*, **Gerd Ahlström**[1], **Magnus Persson**[1], **Lina Behm**[1,3]

1 Faculty of Medicine, Department of Health Sciences, Lund University, Lund, Sweden, 2 Faculty of Health Sciences, Research Platform for Collaboration for Health, Kristianstad University, Kristianstad, Sweden, 3 Faculty of Health Sciences, Kristianstad University, Kristianstad, Sweden

* albert.westergren@med.lu.se

## Abstract

### Background

Next of kin participation in care is a cornerstone of palliative care and is thus important in nursing homes, and outcomes following interventions need to be evaluated using robust methods.

### Objective

To use within-group and within-individual analytical approaches to evaluate the participation of next of kin in care following an intervention and to compare the outcome between the intervention and control groups.

### Methods

A pre–post intervention/control group study design was used. The educational intervention, directed towards staff members, focused on palliative care. The Next of Kin Participation in Care scale comprises the Communication and Trust subscale and the Collaboration in Care subscale, with nine items each. In total, 203 persons (intervention group: n = 95; control group: n = 108) were included. Three different analytical approaches were used: 1) traditional within-group comparison of raw ordinal scores and linearly transformed interval scores; 2) modern within-individual (person-level) interval score comparisons; 3) comparisons between the intervention group and control group based on individual person-level outcomes.

### Results

Within-group comparisons of change revealed no change in any of the groups, whether based on raw or transformed scores. Despite this, significant improvements at the individual

**Data Availability Statement:** Data are available from the Swedish National Data Service (https://

doi.org/10.5878/bknc-4277), and upon request from the first author.

**Funding:** This study is part of the KUPA project and was funded by the Swedish Research Council (Grant number 2014-2759); the Vårdal Foundation (Grant number 2014-0071); the Medical Faculty, Lund University; the Faculty of Health and Life Sciences, Linnaeus University; and the city of Lund. Additionally, a grant from the Foundation of Hedda Andersson, Lund University made this study possible. The authors declare no conflict of interest. The grant sponsors had no role in the design of the study; in the collection, analyses, or interpretation of the data; in the writing of the manuscript or in the decision to publish the results.

**Competing interests:** The authors have declared that no competing interests exist.

level were found in 32.9% of the intervention group and 11.6% of the control group for the total scale (p = 0.0024), in 25% of the intervention group and 10.5% of the control group for the Communication and Trust subscale (p = 0.0018), and in 31.2% of the intervention group and 10.5% of the control group for the Collaboration in Care subscale (p = 0.0016). However, a significant worsening at the individual level in Collaboration in Care was found in 35.1% of the intervention group but only among 8.4% of the control group (p < 0.0005).

## Conclusion

The intervention seems to have a positive impact on next of kin participation in care in nursing homes, especially for communication and trust. However, some next of kin reported decreased participation in care after the intervention. Modern individual person-level approaches for the analysis of intervention outcomes revealed individual significant changes beyond traditional group-level comparisons that would otherwise be hidden. The findings are relevant for future outcome studies and may also necessitate a re-evaluation of previous studies that have not used individual person-level comparisons.

## Trial registration

This study is part of the intervention project registered under Clinical Trials Registration NCT02708498.

## Introduction

The participation of next of kin (NoK) is a cornerstone of high-quality palliative care [1], and as has been expressed in policy documents [2,3] and translated into practice through advanced care planning [4]. However, a previous study found that staff in nursing homes focus on 'getting the work done' rather than on the relationship or communication with NoK [5]. NoK who not have a good relationship to staff members during palliative care have expressed feelings of powerlessness and being left out, which can lead to estrangement and a less good death for the older person [6,7]. Several barriers to communication have been identified in nursing homes, such as NoK role confusion, conflicting responsibilities, understaffing, turnover and inadequate staff training [8]. Inadequate training for staff members is a well-known barrier in the provision of palliative care [9]; however, few initiatives for staff training in palliative care have been implemented, and this is especially true for initiatives focusing on the participation of NoK. A recent scoping review of studies aiming to improve staff members' competence in palliative care with a focus on relationships with NoK found 22 articles describing educational initiatives but only three that focused on NoK relationships [10]. The study concluded that there is a need for further research that explores NoK outcomes using robust methods [10], including robust outcome measures and conducting appropriate statistical comparisons of different groups.

First, in terms of outcome measures, we have previously described the concepts included in the Next of Kin Participation in Care (NoK-PiC) questionnaire, which is designed to measure NoK's participation in care in nursing homes [11]. Specifically, these concepts are trusting the staff, being present, conversations and information, relationships with the staff, completing tasks, being respected for one's knowledge and being acknowledged as part of the care team [11]. It has been found that participation in care can be measured using the NoK-PiC scale,

which includes the Communication and Trust (CaT) and Collaboration in Care (CiC) sub-scales. The development of these two subscales was based on Rasch model analysis (RMA) [12–14]. This article presents results from the first application of the NoK-PiC in an intervention study of palliative care.

Second, in terms of statistical analysis, traditional group comparisons have been found to hide significant outcomes at person level [15]. Hobart et al. [15] stated that 'group-based analyses should be complemented by legitimate analyses at the individual person level' (p. 1048). The scales and analytical approaches used to evaluate the effects of interventions contribute to decisions about which interventions to implement in practice. In this decision-making process, rating scales and comparative statistics play crucial roles in choices that impact clinical practice. This study evaluated an intervention outcome by applying different analytical approaches: traditional within-group comparison of raw ordinal scores; traditional within-group comparison of transformed interval scores and modern within-individual (person-level) interval score comparisons [15]. The same analytical approaches as recommended by Hobart et al. [15] were used in this study to evaluate in detail the effects of the intervention. To the best of our knowledge, no previous intervention studies that had a control group and were based on rigorous outcome measures have explored the participation of NoK in care following an intervention using within-person interval score comparisons. Hobart et al. (2010) empha-sised the clear need for further work using scales that fit the Rasch model requirements to explore significant outcomes at the person level [15]. This study focuses on the NoK-PiC scale as the outcome measure.

The aim of this study was to describe and evaluate the outcome of NoK's participation in care in an intervention and a control group through different analytical approaches (tradi-tional within-group comparison of raw ordinal scores, traditional within-group comparison of transformed interval scores and modern within-individual (person-level) comparisons) after an educational intervention focusing on palliative care for older persons in nursing homes that was directed towards nursing home staff members. A further aim was to compare the interven-tion group and control group based on the individual person-level of change.

## Methods

### Design

This study used a descriptive and comparative evaluation design in a pre–post intervention setting.

### Setting

**The research setting.** This study is part of the 'Implementation of Knowledge-based Palli-ative Care' project, which is abbreviated in Swedish as the KUPA project (*Implementering av KUnskapsbaserad PAlliativ vård*). The KUPA project was a non-randomised experimental two-armed crossover study [16]. A mix of nursing homes participated in the study, including larger (> 100 older persons), middle-sized (25–100 older persons) and smaller (< 25 older persons) facilities located in both urban and rural areas. The project implemented an educa-tional intervention across 30 nursing homes (intervention [n = 10], control [n = 10], and first control and then intervention [n = 10]). The intervention consisted of five 2-hour seminars addressing the knowledge and skills considered necessary to provide evidence-based palliative care in nursing homes. The content was based on fundamental principles of palliative care found in two Swedish documents: a national care programme produced by the Regional Co-operative Cancer Centres [17] and a national knowledge support document produced by the National Board of Health and Welfare [1]. Both of these Swedish documents are based on the

World Health Organization definition of palliative care [2,3,18,19]. The topics covered by each seminar were: 1) the palliative approach and dignified care; 2) NoK; 3) existence and dying; 4) symptom relief and 5) collaborative care [16].

**Sampling and the study group.** The participants in this study were recruited from all 30 nursing homes in the KUPA project, which are located in two counties in the south of Sweden. The inclusion criteria for individual participants were being NoK to an older person living in one of the included nursing homes in the KUPA project and being able to speak and understand Swedish [16]. A total of 203 NoK (n = 95 in the intervention group and n = 108 in the control group) were included. There were no significant differences between the intervention group and the control group at baseline in terms of age, sex or type of relationship to the older person living in the nursing home.

**Data collection.** A contact person (a nurse assistant or a manager) at each of the included nursing homes informed the NoK who fulfilled the inclusion criteria about the study and asked whether they were interested in participating. If they responded positively, the contact person sent the NoK's contact information to the researcher, who then contacted the NoK by telephone, further informed them about the study and asked them whether they consented to participate. When the NoK provided consent, the researchers sent the questionnaire along with a consent form for the baseline assessment via regular mail to the NoK. The researchers then sent a follow-up questionnaire by regular mail 3 months after the completion of the educational seminars (i.e. 9 months after the NoK responded to the first questionnaire).

## Questionnaire

The self-report NoK-PiC questionnaire was developed on the basis of a review of the literature, and some items were inspired by the Family Collaboration Scale [20]. The NoK-PiC scale consists of two subscales, the CaT and the CiC, which can either be used separately or combined to produce a total score. The scales meet rigorous measurement standards, having, for instance, no differential item functioning, high person separation indexes and no disordered thresholds [11]. The subscales contain nine items each, and all items are scored from 0 to 4 (*do not agree at all* [= 0]; *agree to a low extent* [= 1]; *partially agree* [= 2]; *agree to a high extent* [= 3] and *totally agree* [= 4]). Possible scores range from 0 to 36 on each of the two subscales and from 0 to 72 on the total scale [11].

## Data analysis

The percentages of persons with the lowest (floor) and highest (ceiling) scores were calculated. It is recommended that these not exceed 20% [21]. Comparisons were made to explore whether there were differences in baseline characteristics between the intervention group and the control group. These independent group comparisons were made using the chi-square test, the *t*-test and the Mann–Whitney *U* test. IBM SPSS Statistics for Windows, Version 23.0 (IBM Corp., Armonk, NY, USA) was used for these analyses.

In the context of this study, RMA advances the possibilities for evaluating intervention effects. First, RMA allows interval-level (linear) measurements to be estimated from ordinal-level scores. Second, RMA enables the examination of changes at the individual person-level, beyond traditional group-level comparisons [15,22]. Ordinal scores were calculated by summing the score of each item and then transformed into interval-level measurements through RMA. These estimates, termed 'person locations', are in log-odds units (logits). For each person location, the RMA also generated a standard error (SE). The raw score transformation to logits and SEs has previously been presented for the NoK-PiC and its two subscales (the CaT and the CiC) [11]. RMA has been explained in detail elsewhere [12–14]. In this study, the

RMAs were conducted using RUMM2030 Professional Edition 5.4 (RUMM Laboratory Pty Ltd, Duncraig, Australia) [23].

**Traditional within-group comparisons.** Within-group comparisons of total ordinal scores were conducted using the paired samples *t*-test and the Wilcoxon signed-rank test. The ordinal data were normally distributed (skewness and kurtosis were within the range of -1 to +1 [specific range: -0.912 to 0.164]). Comparisons of raw scores that were linearly transformed to person locations (logits) were made using the paired samples *t*-test. Responsiveness, in terms of the magnitude of change over time [24], was estimated using the Kazis effect size (ES = mean change score/standard deviation [SD] of baseline score) [25] and the standardised response mean (SRM = mean change score/SD of change score) [26], following the analytical approaches used by Hobart et al. (2010). Because there is a lack of uniform and widely accepted criteria to give meaning to the size of an effect [24], we chose to use the cut-offs suggested by Cohen [27], although these are based on calculations with the pooled SD [24]. Thus, ES and SRM were interpreted as trivial when the value was < 20, small when the value was $\geq$ 0.20 and < 0.50, moderate when the value was $\geq$ 0.50 and < 0.80, and large when the value was $\geq$ 0.80 [27].

**Modern within-individual (person-level) comparisons.** Within-individual comparisons were conducted based on the approach described in detail by Hobart et al. (2010) and McCarthy et al. (2012) [15,22], and the change at the individual level was assessed by computing, for each person, the significance of the individual's own change in participation in care following the intervention (Sig Change). First, the size of the change for each person was computed (follow-up location − baseline location). Second, the size of the error associated with the change was computed (SE of the difference; SEdiff). Third, the significance of the change for each individual was computed by dividing their change score by their SEdiff. Finally, the significance of each person's change was categorised into one of five groups according to the size and direction of the significance of their change value. The formulae are as follows:

$$\text{Sig Change} = (\text{follow-up location} - \text{baseline location})/\text{SEdiff},$$

where SEdiff = square root [(SE baseline location)$^2$ + (SE follow-up location)$^2$].

The categorisation of the significance of each person's change into five groups was made as follows:

- Significant improvement: Sig Change $\geq$ +1.96
- Non-significant improvement: 0 < Sig Change $\leq$ +1.95
- No change: Sig Change = 0
- Non-significant worsening: -1.95 $\leq$ Sig Change < 0
- Significant worsening: Sig Change $\leq$ -1.96

**Between-group comparisons based on individual person-level outcomes.** Although the main focus of this study was on within-group and within-individual (person-level) comparisons, between-group analyses were also conducted for comparisons between the intervention and control groups. Between-group comparisons were conducted by counting the number of people achieving each level of significance of change, and the distributions were compared using chi-square tests and relative risks (RR). The RR is estimated as the absolute risk in the intervention group divided by the absolute risk in the control group. A value of one indicates no difference in risk, greater than one indicates increased risk, and a value lower than one indicates decreased risk. The RRs and their 95% confidence intervals were calculated using an online resource: https://www.medcalc.org/calc/relative_risk.php. Effect sizes for mean

differences between groups with unequal sample sizes within a pre–post control design were also calculated [28] using an online resource: https://www.psychometrica.de/effect_size. html#cohc. These effect sizes were interpreted as described above [27].

### Ethical considerations

The KUPA project, including this study, has been approved by the Regional Ethics Review Board in Lund, Sweden (no 2015/69), and the project is registered in the ClinicalTrial database for clinical research (NCT02708498). This study was based on informed consent and guided by the ethical principles for medical research in the Declaration of Helsinki [29]. Information provided before the start of the study presented the aim and design of the study, as well as describing participants' right to withdraw from the study at any time without suffering any consequences. Both oral and written informed consent was received from each participant before they responded to the questionnaire. The management of the data is in agreement with the General Data Protection Regulation, and the code lists identifying individual study participants are stored in locked cabinets separate from the questionnaire forms. The participants' confidentiality was respected, and the results have therefore been reported at group level or using non-traceable case numbers.

## Results

In the intervention group (n = 95), 80 (84%) of the participants completed the CaT subscale, 77 (81%) completed the CiC subscale and 70 (74%) completed the total NoK-PiC scale at both baseline and follow-up. The corresponding figures in the control group (n = 108) were 95 (88%) both for the CaT subscale and for the CiC subscale, and 86 (80%) for the total NoK-PiC scale. Participants for whom a total NoK-PiC score could not be computed because of nonresponse to relevant items at baseline and/or follow-up (n = 47) did not differ significantly at baseline from those for whom these scores could be computed (n = 156) in terms of age or gender. There were no significant differences between the intervention and control groups at baseline on CaT, CiC or NoK-PiC raw ordinal scores (Table 1).

### Traditional within-group comparisons

When comparing ordinal raw scores and interval scores between baseline and follow-up, there were no significant changes in any of the scales within the two groups. All ESs and SRMs could be considered trivial (Table 2).

### Modern within-individual (person-level) comparisons

Despite traditional within-group comparisons both at the level of raw ordinal scores and of linearly transformed interval scores revealing no change in any of the groups, significant changes were seen when using individual person-level comparisons. This is illustrated in Fig 1, where individual person-level changes are shown for five cases illustrative of: significant improvement (Panel A); non-significant improvement (Panel B); no change (Panel C); non-significant worsening (Panel D); and significant worsening (Panel E).

### Between-group comparisons based on individual person-level outcomes

When comparing individual person-level outcomes, there was a difference in CaT between the intervention group and the control group (p < 0.0005; Table 3). More specifically, there was a significantly greater chance for improvement in CaT score in the intervention group (25.0%) than in the control group (6.3%, p = 0.002). CiC also differed significantly between the

**Table 1. Sample characteristics at baseline for the intervention and control groups.**

| | Intervention, n = 95 | Control, n = 108 | P-value |
|---|---|---|---|
| **Age**, mean (SD) | 64.99 (9.94) | 64.42 (9.10) | 0.678[a] |
| Gender, women, n (%) | 70 (73.7) | 86 (79.6) | 0.316[b] |
| **Relationship**, n (%) | | | 0.979[b] |
| Husband/wife | 19 (20.0) | 20 (18.9) | |
| Daughter/son, stepdaughter/stepson | 69 (72.6) | 78 (73.6) | |
| Other | 7 (7.4) | 8 (7.5) | |
| **Communication and Trust subscale (CaT)**, baseline ordinal score, mean (SD) | 27.57 (7.21) | 27.96 (6.62) | 0.744[c] |
| Floor effect, n (%) | 0 | 0 | - |
| Ceiling effect, n (%) | 7 (7.9) | 9 (9.0) | - |
| **Collaboration in Care subscale (CiC)**, baseline ordinal score, mean (SD) | 21.38 (8.63) | 20.68 (8.78) | 0.702[c] |
| Floor effect, n (%) | 0 | 0 | - |
| Ceiling effect, n (%) | 3 (3.2) | 6 (5.9) | - |
| **Next of Kin Participation in Care total scale (NoK-PiC)**, baseline, ordinal score, mean (SD) | 48.95 (14.87) | 48.64 (14.09) | 0.775[c] |
| Floor effect, n (%) | 0 | 0 | - |
| Ceiling effect, n (%) | 3 (3.6) | 5 (5.3) | - |

Up to 20% floor/ceiling effects were considered acceptable [21].

[a]Independent samples *t*-test.

[b]Chi-square test.

[c]Mann–Whitney *U* test.

intervention group and the control group (p < 0.0005), in both the percentage experiencing significant improvement (31.2% vs. 10.5%, p = 0.002) and the percentage experiencing significant worsening (35.1% vs. 8.4%, p < 0.0005). There was also a significant difference in the total NoK-PiC scale between the intervention group and the control group (p < 0.0005). A larger percentage of persons in the intervention group (32.9%) than in the control group (11.6%) experienced significant improvement between baseline and follow-up (p = 0.002; Table 3).

All ES values between the intervention group and the control group were trivial/small (for ordinal/interval scores respectively, 0.40/0.11 for CaT, -0.11/-0.06 for CiC and 0.02/0.06 for NoK-PiC).

## Discussion

Based on the significant results on the individual level, the intervention can be regarded as successful within the area of CaT, although the success regarding CiC is more doubtful. However, two contradicting conclusions can be reached depending on which analytical approach is chosen for the analysis of the data and interpretation of the results. First, the ordinal and interval scores showed no change in participation within any of the groups. Second, considering individual person-level change, both significant improvement as well as worsening were revealed in both the intervention group and the control group. In between-group comparisons, there was an improvement in participation with respect to both CaT and CiC in the intervention group, compared with the control group. However, there was also a higher risk of significant worsening in CiC in the intervention group than in the control group. For the total NoK-PiC scale, there was a greater individual person-level positive change in the intervention group than in the control group.

This study has three major findings that need to be discussed. The first relates to whether or not the intervention can be regarded as successful. The second relates to the diverse findings

**Table 2.  Communication and Trust (CaT), Collaboration in Care (CiC) and Next of Kin Participation in Care (NoK-PiC) baseline and follow-up of ordinal and interval scores.**

| | Intervention | | | Control | | |
|---|---|---|---|---|---|---|
| | CaT | CiC | NoK-PiC | CaT | CiC | NoK-PiC |
| **Ordinal scores** | | | | | | |
| Possible range | 0–36 | 0–36 | 0–72 | 0–36 | 0–36 | 0–72 |
| Baseline | | | | | | |
| Mean | 25.6 | 21.4 | 48.9 | 28.0 | 20.7 | 48.6 |
| SD | 7.2 | 8.6 | 14.9 | 6.6 | 8.8 | 14.1 |
| Follow-up | | | | | | |
| Mean | 27.7 | 21.0 | 48.7 | 27.3 | 21.3 | 48.7 |
| SD | 6.7 | 8.5 | 14.3 | 7.2 | 9.3 | 15.4 |
| Change | | | | | | |
| Mean | 0.5 | -0.3 | 0.1 | -0.5 | 0.3 | -0.2 |
| SD | 9.8 | 11.5 | 19.9 | 4.0 | 5.0 | 8.3 |
| Paired samples *t*-test | | | | | | |
| p-value | 0.515 | 0.968 | 0.957 | 0.115 | 0.107 | 0.795 |
| Wilcoxon signed-rank test | | | | | | |
| p-value | 0.379 | 0.989 | 0.950 | 0.129 | 0.120 | 0.765 |
| ES | 0.07 | -0.03 | 0.01 | -0.08 | 0.03 | -0.01 |
| SRM | 0.05 | -0.03 | 0.00 | -0.12 | 0.06 | -0.02 |
| **Interval scores** | | | | | | |
| Possible range[a] | -2.1–5.6 | -3.6–4.9 | -2.0–5.7 | -1.8–5.6 | -3.6–4.9 | -2.4–5.7 |
| Baseline | | | | | | |
| Mean | 2.3 | 0.6 | 1.3 | 2.4 | 0.5 | 1.3 |
| SD | 1.9 | 1.7 | 1.6 | 1.8 | 1.8 | 1.7 |
| Follow-up | | | | | | |
| Mean | 2.4 | 0.6 | 1.4 | 2.3 | 0.6 | 1.3 |
| SD | 1.9 | 1.8 | 1.7 | 1.9 | 1.8 | 1.7 |
| Change | | | | | | |
| Mean | 0.1 | -0.0 | 0.1 | -0.1 | 0.0 | -0.0 |
| SD | 2.6 | 2.3 | 2.2 | 1.2 | 1.0 | 1.0 |
| Paired samples *t*-test | | | | | | |
| p-value | 0.530 | 0.931 | 0.753 | 0.213 | 0.140 | 0.862 |
| ES | 0.05 | -0.06 | 0.06 | -0.05 | 0.05 | -0.06 |
| SRM | 0.04 | -0.04 | 0.04 | -0.08 | 0.10 | -0.01 |

ES: Kazis effect size = mean change score/SD of baseline score. SRM: Standardised response mean = mean change score/SD of change score.

[a]These numbers range from negative to positive values because the person locations are transformed into log-odds units (logits) centred around a mean of zero.

resulting from using different analytical approaches for comparison. The third relates to the conflicting findings regarding the CiC subscale.

First, when focusing on the individual level of change, the intervention can be regarded as successful for CaT but not as successful for CiC. This finding was somewhat expected because it has been hypothesised that it is easier for care interventions to achieve success in improving CaT than in improving CiC [11]. Previous research has shown somewhat similar results. For example, Maas and colleagues [30] evaluated an intervention including 8 hours of training for staff members over three sessions to promote family participation in nursing homes. The study showed significant improvements in family members' perceptions of relationships with the staff [30]; however, no differences were found concerning their perceptions of partnership.

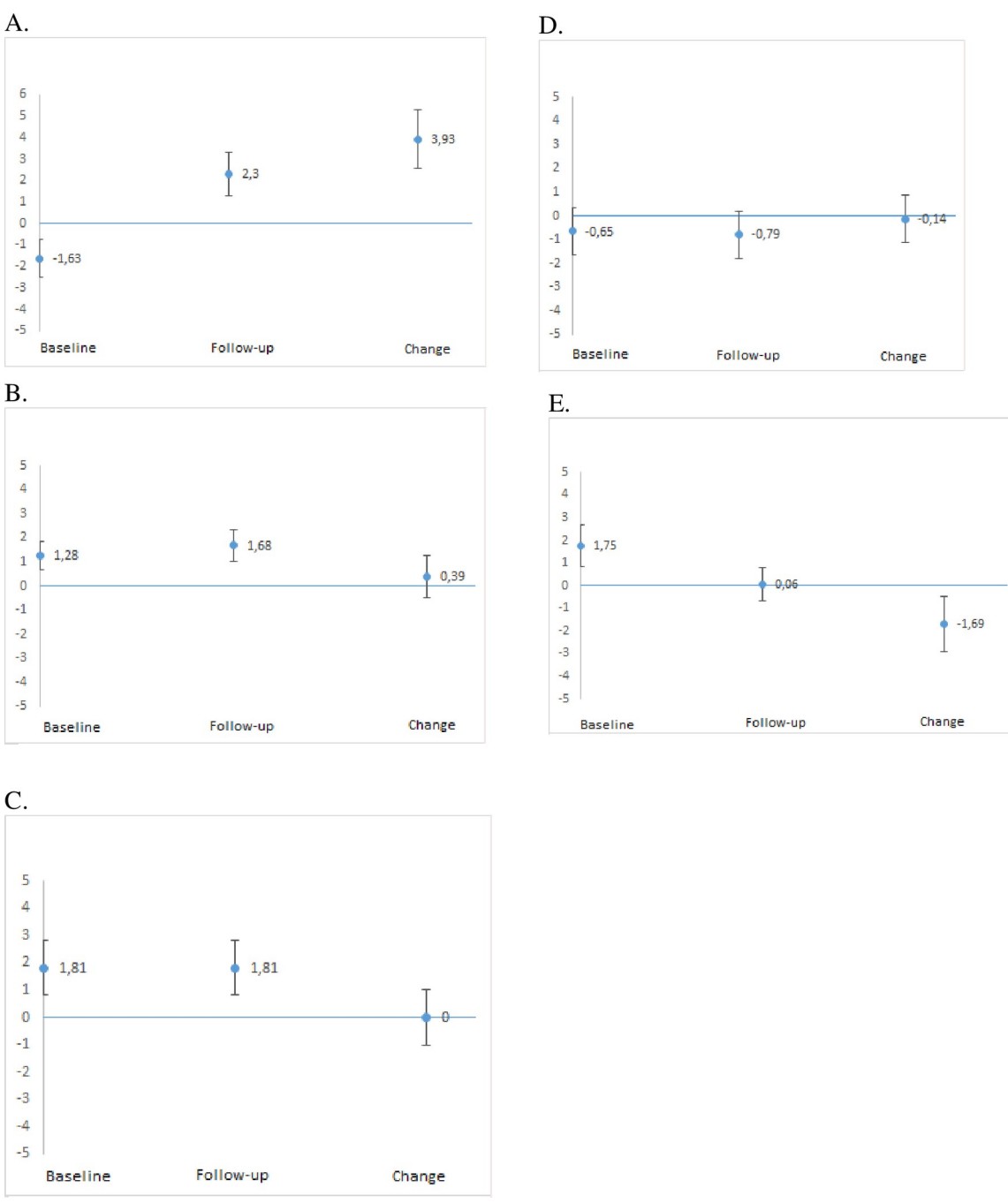

**Fig 1. Individual person-level raw total scores translated to linear logit scale location values (y-axis) for baseline, follow-up, and change in locations (follow-up location – baseline location) (x-axis) together with 95% confidence intervals (CIs).** Panel A illustrates significant improvement (95% CI for change does not overlap zero) in Communication and Trust (CaT) score for Case 147. Panel B illustrates non-significant improvement in Next of Kin Participation in Care (NoK-PiC) score for Case 25. Panel C illustrates no change in CaT score for Case 186. Panel D illustrates non-significant worsening in Collaboration in Care (CiC) score for Case 13. Panel E illustrates significant worsening (95% CI for change does not overlap zero) in CiC score for Case 179.

Another study, conducted by Beck and colleagues [31], used study circles about palliative care for nurse assistants to implement a palliative care approach in nursing homes. The qualitative study showed that the nurse assistants experienced a deeper understanding of NoK's needs

**Table 3. Comparisons of individual person-level (IPL) change in Communication and Trust, Collaboration in Care subscale scores, and Next of Kin Participation in Care (NoK-PiC) scores between the intervention group and the control group.**

| IPL significance of change[a] | Intervention n (%) | Control n (%) | P-value[b] | RR (95% CI) | RR P-value |
|---|---|---|---|---|---|
| **Communication and Trust (CaT)** | n = 80 | n = 95 | < 0.0005 | | |
| Significant improvement | 20 (25.0) | 6 (6.3) | | 3.96 (1.67–9.38) | 0.002[c] |
| Non-significant improvement | 21 (26.3) | 28 (29.5) | | 0.89 (0.55–1.44) | 0.637 |
| No change | 8 (10.0) | 14 (17.5) | | 0.68 (0.30–1.53) | 0.352 |
| Non-significant worsening | 14 (17.5) | 37 (38.9) | | 0.45 (0.26–0.77) | 0.004 |
| Significant worsening | 17 (21.3) | 10 (10.5) | | 2.02 (0.98–4.16) | 0.057 |
| **Collaboration in Care (CiC)** | n = 77 | n = 95 | < 0.0005 | | |
| Significant improvement | 24 (31.2) | 10 (10.5) | | 2.96 (1.51–5.81) | 0.002[c] |
| Non-significant improvement | 16 (20.8) | 38 (40.0) | | 0.52 (0.31–0.86) | 0.010 |
| No change | 2 (2.6) | 8 (8.4) | | 0.31 (0.676–1.41) | 0.129 |
| Non-significant worsening | 8 (10.4) | 31 (32.6) | | 0.32 (0.15–0.65) | 0.002[c] |
| Significant worsening | 27 (35.1) | 8 (8.4) | | 4.16 (2.01–8.64) | < 0.0005[c] |
| **Next of Kin Participation in Care (NoK-PiC)** | n = 70 | n = 86 | < 0.0005 | | |
| Significant improvement | 23 (32.9) | 10 (11.6) | | 2.83 (1.44–5.53) | 0.002[c] |
| Non-significant improvement | 15 (21.4) | 31 (36.0) | | 0.59 (0.35–1.01) | 0.054 |
| No change | 1 (1.4) | 7 (8.1) | | 0.17 (0.02–1.39 | 0.100 |
| Non-significant worsening | 7 (10.0) | 26 (30.2) | | 0.33 (0.15–0.72) | 0.005 |
| Significant worsening | 24 (34.3) | 12 (14.0) | | 2.46 (1.33–4.55) | 0.004 |

RR: Relative risk; CI: Confidence interval.

[a]Significance of change = Follow-up location–Baseline location/SE of the difference.

[b]Chi-square test.

[c]Significant after Bonferroni correction = p < 0.003 (0.05/15).

and stressed the importance of including NoK in the care if they wished to participate [31]. However, the study did not evaluate whether this was actually done in practice, from the NoK's perspectives.

The second important result from the present study relates to the diverse findings when using different analytical approaches for comparisons. Depending on whether we relied on comparisons of ordinal scores, interval scores or individual person-level significant change, we reached different conclusions. This finding raises several questions: What analytical approaches should be used for evaluating the effects of interventions? What are the implications of different analytical approaches for evaluating intervention outcomes in future studies? Are current ordinal score-based standards for evaluating intervention effects, including meta-analysis, not the best path forward? It might be that we need to develop a more individual, person-based evaluation of interventions. As illustrated in this study, this approach can be achieved through Rash measurement-based criteria. Thus, the findings from this study might have implications for the assessment of previous outcome evaluation research and for the analysis conducted in future research. It is possible that many successful interventions have not been discovered because of the predominance of a group-level focus on ordinal score significance rather than a person-centred focus on individual and interval level-based significance. However, we do not have a full explanation for our findings, and there is thus a clear need for further work in line with the analytical approaches described here and in previous work conducted by Hobart et al. 'to elaborate upon what we have uncovered here and to ultimately pin down its root cause' [15] (p. 1047). The findings from the present study indicate that standard group-level analyses are limited and possibly misleading.

Since individual person-level analytical approaches can unpick the nuances beyond aggregated group-level data, it becomes possible to find out to what extent an intervention is effective, in other words, what works, for whom and in what circumstances. Traditional evaluation efforts that focus on aggregate effectiveness have been criticized for representing an oversimplification [32,33]. Through individual person-level analytical approaches we can begin to explore why interventions worked for certain persons and not others. This is in line with the recommendation by Pawson and Tilley [34], developers of the realist evaluation approach, to conduct evaluations for subgroups within programmes; they advised researchers to be cautious since there might be more than one mechanism at work within each subgroup, generating mixed results [34], which was also revealed by the findings in this study. Realist evaluation is situated between positivism and realism [35], and it attempts to explore contextual circumstances in which mechanisms are triggered and lead to outcomes [34]. Interestingly though, many realist evaluations, although being neutral on the qualitative-quantitative spectrum, tend to be small-scale, mixed-method or qualitative case studies. One criticism is that they lack generalisability beyond the case study unit of analysis, and sometimes the evaluations are unclear about context, mechanism and/or outcome [32,36–38]. The area of outcome assessment tends to be especially problematic [32]. Taken together, these issues can make it hard for realist evaluations to gain scientific credibility [39]. However, this is a point for debate as realism, for some, is not necessarily congruent with generalisability, i.e. neither from a positivism nor an interpretivism point of view [40]. It has been stated that "Prevailing statistical models which by their nature are aggregate /.../ may have limited utility in the analysis of complex systems" [33] (p 388). We claim that modern individual person-level analytical approaches may support larger scale realist evaluations as part of a mixed-methods study design.

The third important finding relates to the diverse results regarding CiC. In the intervention group, compared with the control group, there were significantly higher numbers of both persons with significant improvement and persons with significant worsening in terms of CiC. CiC measures collaboration, meaning involving more action from the NoK's perspective, in contrast to the CaT subscale, which focuses more on the prerequisites for participation. NoK involvement in care results from both the staff members' expectations and the NoK's will/ability to get involved. Correspondingly, because NoK functioning as actors in care must be voluntary, it can be more difficult for interventions to achieve high scores on the CiC scale than on the CaT scale [11]. Although many NoK want to participate in the care of their relatives [41], others experience pressure to take on more tasks than they wish [41,42]. Thus, it is possible that staff members express a desire to involve the NoK in care as a 'requested' indirect side effect of the intervention. This may be welcomed by the NoK, leading to an increase in involvement, but it may also be the case that they expect the staff to involve them more than actually occurs, leading to a decrease in the rating of participation as measured by the CiC scale. Thus, NoK participation in care is a balancing act between NoK maintaining their own responsibility while also ceding responsibility to the nursing home staff [43]. Our findings indicate a possible imbalance between staff expectations and NoK's will/ability developing over time as a side effect of educational interventions for staff members.

A first methodological aspect of the present study that needs to be discussed relates to responsiveness and ES. Scale responsiveness and treatment effectiveness are inseparably linked. ES, computed using change in scores from baseline to follow-up, is an indicator of both the ability of a scale to detect change and the magnitude of the intervention effect [24,44]. There is, however, a lack of consensus on the most appropriate effect size indicator to use [24]. Further, in matched-pair studies, the cut-offs suggested by Cohen (1977) cannot be used interchangeably for the SRM because of the correlation between scores within pairs [24]. However, because we used ES and SRM mainly for explorative purposes and because low ES was

expected, we followed the interpretation suggested by Cohen (1977); this was also necessary because no substantiated alternative exists for the interpretation of ES. SRM was calculated in addition to ES to determine whether the findings were specific to one computation, and this was not found to be the case.

A second methodological aspect relates to the understanding of meaningful change from a perspective 'outside' the individual level and from an 'inside' person-level perspective. It should be noted that, although ES has been used as a proxy for clinically important change, this measure does not provide a complete understanding of the meaningfulness of the observed change. What does the change mean for the respondent? A respondent may perceive even a small change as significant [24], but this might not be captured by ES or other traditional analytical approaches for group comparison. This aspect of the subjective perception of important change is of course also relevant for the individual person-level of significance. Thus, there seems to be a need for approaches that take more explicit account of the person's own perceptions of improvement or worsening. Individuals' preferences are thus essential pieces of information. RMA offers a pathway towards individual person-level analysis of intervention outcomes, especially if it is based on person-centred outcome measures focusing on what individuals themselves perceive as important.

## Conclusions

Can an educational intervention for staff members focusing on palliative care be recommended for increasing NoK's participation in care? The answer is both yes and no. If the goal is to improve CaT, the answer is yes. If the goal is also to improve NoK's CiC, the results of this study provide no clear answer.

Different conclusions regarding intervention outcomes can be drawn, depending on which analytical approaches are used for comparisons. Modern individual person-level methods for analysis of intervention outcomes uncover individual significance that cannot be detected by traditional group-level comparisons and would thus otherwise be hidden. The findings of our study have the potential to impact future outcome studies and may also necessitate a re-evaluation of previous studies of this type.

## Acknowledgments

We thank the researchers at Lund University who participated in the data collection: Birgitta Wallerstedt (PhD, RN), Helene Åvik Persson (PhD student, PHN) and Anne Molina Tall (research assistant, RN). We would also like to acknowledge the cooperation of Birgit Rasmussen (professor) and Tove Lindhardt (RN, PhD). We also thank Jennifer Barrett, PhD, from Edanz Group (https://en-author-services.edanzgroup.com/), and Catriona Chaplin, PhD, CMC Scientific English for Publication (https://scientificenglish.se/), for editing a draft of this manuscript.

## Author Contributions

**Conceptualization:** Albert Westergren.

**Data curation:** Gerd Ahlström, Magnus Persson, Lina Behm.

**Formal analysis:** Albert Westergren.

**Funding acquisition:** Gerd Ahlström.

**Investigation:** Albert Westergren.

**Project administration:** Gerd Ahlström, Magnus Persson.

**Resources:** Gerd Ahlström.

**Software:** Albert Westergren.

**Validation:** Albert Westergren, Gerd Ahlström, Magnus Persson, Lina Behm.

**Visualization:** Albert Westergren.

**Writing – original draft:** Albert Westergren, Gerd Ahlström, Magnus Persson, Lina Behm.

**Writing – review & editing:** Albert Westergren, Gerd Ahlström, Magnus Persson, Lina Behm.

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
