## [Decision Letter · Decision Letter 0]

22 Oct 2020

PONE-D-20-16638

Evaluating participation of next of kin after an educational intervention in nursing homes using methods for individual person-level comparison

PLOS ONE

Dear Dr. Westergren,

Thank you for submitting your manuscript to PLOS ONE. After careful consideration, we feel that it has merit but does not fully meet PLOS ONE’s publication criteria as it currently stands. Therefore, we invite you to submit a revised version of the manuscript that addresses the points raised during the review process.

Please review the recommendations and suggestions from the reviewers. In particular please attend to the title and abstract to ensure they adequately reflect the main messages and methods of the paper, and revise the Methods, Results and Discussion sections to ensure consistency.  

We look forward to receiving your revised manuscript.

Kind regards,

Lucy Ellen Selman, BA, MPhil, PhD

Academic Editor

PLOS ONE

Journal Requirements:

Reviewers' comments:

Reviewer's Responses to Questions

**Comments to the Author**

1. Is the manuscript technically sound, and do the data support the conclusions?

Reviewer #1: Yes

Reviewer #2: Yes

2. Has the statistical analysis been performed appropriately and rigorously? 

Reviewer #1: Yes

Reviewer #2: Yes

3. Have the authors made all data underlying the findings in their manuscript fully available?

Reviewer #1: Yes

Reviewer #2: Yes

4. Is the manuscript presented in an intelligible fashion and written in standard English?

Reviewer #1: Yes

Reviewer #2: Yes

5. Review Comments to the Author

Reviewer #1: Overall, the manuscript was concise and comprehensive. The author uses two statistical analyses on the data, group, and individual, arriving at different conclusions. This was a unique method to demonstrate significance in alternative results. Also, the author provides explanations as to why results may not have been significant (e.g., next of kin not wanting to take on additional care). However, it may have been beneficial if the author focused on the individual changes instead of comparing group to individual results. It was challenging to determine which overall findings the reader should accept. If this was a psychometric study, it needs to be clearly stated at the beginning of the manuscript.

Reviewer #2: Thank for you inviting me to review this manuscript, which makes the case for individual person-level methods for analysis in the context of an educational intervention in nursing homes.

Generally, I found this manuscript to be well written and interesting. There are, however, a couple of exceptions in respect of the clarity of the manuscript.

1) Upon first reading the title and abstract, I didn’t feel that they adequately describe the paper. For example, it is not clear from the abstract (see objective) what the three methods are, and I think they may be better described as analytical approaches. In my view, a method refers to a means of data collection (e.g. interview or survey), whereas I think the authors are talking about analytical approaches.

2) The three analytical approaches should be described before the Results section, given this is the key focus of the paper. This only becomes clear in the results and conclusion, and even then not introducing it earlier makes the key focus of the paper unclear in the abstract.

3) I think the authors should consider naming the three analytical approaches in the title, and then briefly mention and describe in the background and objective sections. This will then mean the manuscript has an obvious and clear focus.

4) Under Trial registration it says that the manuscript is a psychometric study – but this is the first time it is mentioned. Something perhaps also to consider adding to the title and abstract.

As an active researcher in this field I found the rest of the paper to be of significant interest, particularly the development of the new scales/outcome measures (which I may even use myself at some point).

One additional point to consider including in the discussion is whether this manuscript can feed into the relatively emerging field of realist methodology and methods. Because the researchers highlight that individual person-level methods of analysis can unpick the nuances behind aggregate/sample level data, my interpretation of this is that this analytical approach is capable of going beyond finding out whether an intervention is effective (as denoted by sample level reporting) and could highlight how it might, for whom and in what circumstances – based on being able to begin to explore why interventions worked for certain people, and not others. This is what realist evaluation and realist research attempts to do (by exploring the contextual circumstances in which mechanisms fire and lead to outcomes). Interestingly though, many realist evaluations tend to be small scale mixed method or qualitative case studies, and one criticism is that they lack generalisability beyond the case study/unit of analysis (though this is a point for debate as realism, for many, is not necessarily congruent with generalisability. There are many strands of realist philosophy!). When reading lines 346-364 I am left wondering whether this manuscript may be capable of adding some interesting discussion to the emerging debates on realist research, particularly in identifying an analytical approach that may support larger scale realist evaluations (as part of a mixed methods study design).

6. PLOS authors have the option to publish the peer review history of their article (what does this mean?). If published, this will include your full peer review and any attached files.

Reviewer #1: No

Reviewer #2: No

---

## [Author Response · Author response to Decision Letter 0]

26 Nov 2020

Reply to reviewers 

We wish to thank the academic editor and reviewers for your positive feedback and relevant suggestions for revisions. Below we respond to each point raised.

Academic Editor

Please review the recommendations and suggestions from the reviewers. In particular please attend to the title and abstract to ensure they adequately reflect the main messages and methods of the paper, and revise the Methods, Results and Discussion sections to ensure consistency. 

Response: Thank you for this advice. We have now made changes according to the reviewers’ recommendations, regarding the title, Abstract, Methods, Results and Discussion. 

Reviewer #1

R1: Overall, the manuscript was concise and comprehensive. The author uses two statistical analyses on the data, group, and individual, arriving at different conclusions. This was a unique method to demonstrate significance in alternative results. Also, the author provides explanations as to why results may not have been significant (e.g., next of kin not wanting to take on additional care). 

Response: Thank you for these positive remarks.

R1:Q1: However, it may have been beneficial if the author focused on the individual changes instead of comparing group to individual results. It was challenging to determine which overall findings the reader should accept. 

Response: Instead of deleting the between-group comparisons, which add valuable information when interpreting the findings, we have now clarified that the within-group and individual person-level comparisons are the primary findings. We have also clarified this by adding matching subheadings in the Methods and Results sections. For example (from the Data analysis section):

Between-group comparisons based on individual person-level outcomes

Although the main focus of this study was on within-group and within-individual (person-level) comparisons, between-group analyses were also conducted for comparisons between the intervention and control groups.

R1:Q2: If this was a psychometric study, it needs to be clearly stated at the beginning of the manuscript.

Response: Thank you for pointing this out. We have now deleted “psychometric” under the heading “Trial registration”. 

Reviewer #2

R2: Thank for you inviting me to review this manuscript, which makes the case for individual person-level methods for analysis in the context of an educational intervention in nursing homes. Generally, I found this manuscript to be well written and interesting. There are, however, a couple of exceptions in respect of the clarity of the manuscript. 

Response: Thank you for your positive feedback.

R2:Q1: Upon first reading the title and abstract, I didn’t feel that they adequately describe the paper. For example, it is not clear from the abstract (see objective) what the three methods are, and I think they may be better described as analytical approaches. In my view, a method refers to a means of data collection (e.g. interview or survey), whereas I think the authors are talking about analytical approaches.

Response: Thank you for pointing out this terminology issue. We have now used “analytical approach” instead of “methods”. 

R2:Q2: The three analytical approaches should be described before the Results section, given this is the key focus of the paper. This only becomes clear in the results and conclusion, and even then, not introducing it earlier makes the key focus of the paper unclear in the abstract.

Response: We are grateful to you for this observation. We have now clarified the analytical approaches in the Abstract, Introduction, Methods and Results, for instance by using subheadings. 

R2:Q3: I think the authors should consider naming the three analytical approaches in the title, and then briefly mention and describe in the background and objective sections. This will then mean the manuscript has an obvious and clear focus.

Response: Thank you for this suggestion. To clarify the focus, we have now included these approaches in the title and in various places in abstract, introduction, methods and in results (see also R2:Q2).

R2:Q4: Under Trial registration it says that the manuscript is a psychometric study – but this is the first time it is mentioned. Something perhaps also to consider adding to the title and abstract.

Response: Thank you for pointing this out. We have now deleted “psychometric” under the heading “Trial registration”, since we do not consider this paper to represent a psychometric study. However, the paper preceding this one was a psychometric study.

R2: As an active researcher in this field I found the rest of the paper to be of significant interest, particularly the development of the new scales/outcome measures (which I may even use myself at some point). 

Response: We are delighted that you found the paper relevant and interesting.

R2:Q5: One additional point to consider including in the discussion is whether this manuscript can feed into the relatively emerging field of realist methodology and methods /…/.

Response: This was an excellent idea! We have added the following paragraph about this in the Discussion: 

Since individual person-level analytical approaches can unpick the nuances beyond aggregated group-level data, it becomes possible to find out to what extent an intervention is effective, in other words, what works, for whom and in what circumstances. Traditional evaluation efforts that focus on aggregate effectiveness have been criticized for representing an oversimplification [32, 33]. Through individual person-level analytical approaches we can begin to explore why interventions worked for certain persons and not others. This is in line with the recommendation by Pawson and Tilley [34], developers of the realist evaluation approach, to conduct evaluations for subgroups within programmes; they advised researchers to be cautious since there might be more than one mechanism at work within each subgroup, generating mixed results [34], which was also revealed by the findings in this study. Realist evaluation is situated between positivism and realism [35], and it attempts to explore contextual circumstances in which mechanisms are triggered and lead to outcomes [34]. Interestingly though, many realist evaluations, although being neutral on the qualitative-quantitative spectrum, tend to be small-scale, mixed-method or qualitative case studies. One criticism is that they lack generalisability beyond the case study unit of analysis, and sometimes the evaluations are unclear about context, mechanism and/or outcome [32, 36-38]. The area of outcome assessment tends to be especially problematic [32]. Taken together, these issues can make it hard for realist evaluations to gain scientific credibility [39]. However, this is a point for debate as realism, for some, is not necessarily congruent with generalisability, i.e. neither from a positivism nor an interpretivism point of view [40]. It has been stated that “Prevailing statistical models which by their nature are aggregate /…/ may have limited utility in the analysis of complex systems” [33](p 388). We claim that modern individual person-level analytical approaches may support larger scale realist evaluations as part of a mixed-methods study design.

---

## [Editor Report · Decision Letter 1]

14 Dec 2020

Next of kin participation in the care of older persons in nursing homes: a pre–post non-randomised educational evaluation, using within-group and individual person-level comparisons

PONE-D-20-16638R1

Dear Dr. Westergren,

We’re pleased to inform you that your manuscript has been judged scientifically suitable for publication and will be formally accepted for publication once it meets all outstanding technical requirements.

Kind regards,

Lucy Ellen Selman, BA, MPhil, PhD

Academic Editor

PLOS ONE
---

## [Editor Report · Acceptance letter]

15 Jan 2021

PONE-D-20-16638R1 

Next of kin participation in the care of older persons in nursing homes: a pre–post non-randomised educational evaluation, using within-group and individual person-level comparisons 

Dear Dr. Westergren:

I'm pleased to inform you that your manuscript has been deemed suitable for publication in PLOS ONE. Congratulations! Your manuscript is now with our production department. 

Kind regards, 

on behalf of

Dr. Lucy Ellen Selman 

Academic Editor

PLOS ONE